# QUERY-GUIDED PROTOTYPE GENERATION FOR FEW-CLASS CLASSIFICATION

## ABSTRACT

Recent studies in few-class regime show that the performance of standard image classifiers, especially those trained on many classes, can be significantly degraded when applied to tasks with only a few target categories. In this setting, larger backbones do not necessarily yield better results, and traditional scaling laws often break down, leading to high variance and unpredictable behavior. To address these challenges, we propose a simple yet effective classification module, including prototype generation via query-guided support retrieval and fusion, which can be attached to any frozen image encoder. For each query, a small class-wise support set is retrieved from the training data based on feature similarity to the query. Each retrieved support set is then fused with the query using a transformer module to produce contextual prototypes, which are subsequently processed by a second transformer-based classifier, in which the query attends to the contextual prototypes to produce the final prediction. This approach addresses run-time and memory constraints by restricting attention to a compact set of query-specific prototypes, rather than processing full support sets jointly. It requires no fine-tuning or retraining of the backbone encoder and is compatible with a wide range of architectures. Evaluated across diverse datasets and models from the Few-Class Arena benchmark, our method consistently improves performance over strong baselines and outperforms recent meta-learning methods tailored to this setting. By transforming frozen encoders into query-guided prototype matchers, our approach provides a practical, scalable, and state-of-the-art solution for few-class classification.

## 1 INTRODUCTION

Modern image classifiers have achieved remarkable success in large-scale benchmarks such as ImageNet (Deng et al., 2009b) and COCO (Lin et al., 2014), where hundreds or thousands of classes are used for training and evaluation. Recent advances in vision architectures, including convolutional networks (He et al., 2016; Howard et al., 2019), vision transformers (Dosovitskiy et al., 2020; Liu et al., 2022), and multimodal encoders (Radford et al., 2021; Oquab et al., 2023), have further pushed the performance envelope. These models are typically trained once on massive datasets and then deployed as general-purpose feature extractors or fine-tuned for downstream tasks.

However, real-world applications rarely require the classification of hundreds of categories. Instead, deployed systems often need to distinguish between only a handful of relevant categories, such as classifying medical conditions from scans, detecting a few types of defects in manufacturing, or identifying specific product types in a warehouse. This emerging regime, known as the *Few-Class Regime*, poses new challenges for model evaluation and design. When applied to a small label space, standard classifiers trained on large datasets tend to under-perform, exhibit high variance, and violate traditional scaling laws (Cao et al., 2025b).

Few-shot learning (Vinyals et al., 2016; Snell et al., 2017; Sung et al., 2018) is a meta-learning paradigm aimed at generalizing across tasks. Models are trained on a diverse collection of small classification tasks and evaluated on new tasks with novel classes. Most early methods operate with a small, fixed number of classes per task and assume access to a limited support set of labeled examples. These support sets are typically sampled randomly per class and used uniformly across queries, without regard to their relevance to the specific instance being classified. Embeddings for support

and query examples are computed independently and compared using a fixed-distance metric such as Euclidean or cosine similarity (Vinyals et al., 2016; Snell et al., 2017). Later methods introduced task-conditioned adaptation through gradient-based updates (Finn et al., 2017), and more recent models such as CAML (Fifty et al., 2024a) used transformers to jointly encode the support and query sets, allowing task-level context to emerge through self-attention.

However, these approaches still face important limitations. First, the support sets in prior work are selected independently of the query and sampled uniformly per class, without considering which examples may be most relevant for a given instance. In contrast, we dynamically construct a set of support for each query by retrieving the top k most similar training examples per class, producing a class-balanced context of relevant samples for each query.

Second, methods that do not use prototypes, such as CAML (Fifty et al., 2024a), which take care of the entire support-query sequence, introduce significant memory and compute overhead, particularly as the number of classes and support examples increases.

Third, while some methods aggregate support examples into class prototypes (Snell et al., 2017), these prototypes are computed independently of the query. To our knowledge, we are the first to generate class prototypes by explicitly fusing the query with each class's support examples, resulting in compact, query-aware representations that preserve task-specific context early in the pipeline.

We propose a hybrid framework that combines the scalability of pre-trained image encoders with the adaptability of contextual prototypes generated using query-aware dynamic support sets. Our contributions are as follows:

- Dynamically selecting top-k similar support training samples per query per class.
- A transformer module that fuses each class's support examples with the query to form a query-aware prototype.
- Integrating principles from few-shot learning, including support-based classification using ELMES label embeddings (Fifty et al., 2024a), into a flexible classification module compatible with standard image encoders.
- We demonstrate state-of-the-art performance across multiple datasets from the Few-Class Arena benchmark, surpassing both fine-tuned and sub-model baselines in accuracy and robustness, as well as few-shot methods such as CAML (Fifty et al., 2024a).

## 2 RELATED WORK

Modern visual classifiers are trained on massive benchmarks with hundreds or thousands of categories, driving the development of ever more scalable architectures. Convolutional backbones such as ResNet (He et al., 2016), EfficientNetV2 (Tan & Le, 2021), and MobileNetV3 (Howard et al., 2019) strike different trade-offs between accuracy and efficiency, while transformer-based encoders such as ViT (Dosovitskiy et al., 2021) and Swin Transformer (Liu et al., 2021; 2022) excel at modeling long-range interactions. Self-supervised approaches such as DINOv2 (Oquab et al., 2023) further close the gap to supervised pre-training by learning rich representations without class labels. However, when deployed in small label spaces, these models can suffer unpredictable performance drops.

Cao et al. (2025a) introduced the Few-Class Arena (FCA) benchmark to study this phenomenon explicitly, evaluating standard classifiers in subsets of 2-10 classes drawn from data sets. They show that class-specific submodels often outperform full models and that scaling laws break down in the few-class regime. Several domain-specific studies report similar limitations in the few-class regime. In histopathology (Singh et al., 2021), clinical imaging with foundation models (Woerner & Baumgartner, 2024), and biomedical classification tasks such as those in the MedMNIST v2 suite (Yang et al., 2022), improvements remain modest. Methodological efforts, including non-negative subspace feature learning (Fan et al., 2024), also struggle to produce substantial gains. These findings collectively highlight the need for improved architectural and inference strategies explicitly designed for the challenges of few-class classification.

Few-shot learning (FSL) refers to the problem of generalizing to novel categories from only a handful of labeled examples, typically without any additional fine-tuning at test time. Metric-based methods, such as prototypical networks (Snell et al., 2017), matching networks (Vinyals et al., 2016), and relation networks (Sung et al., 2018), compute class prototypes using randomly selected support

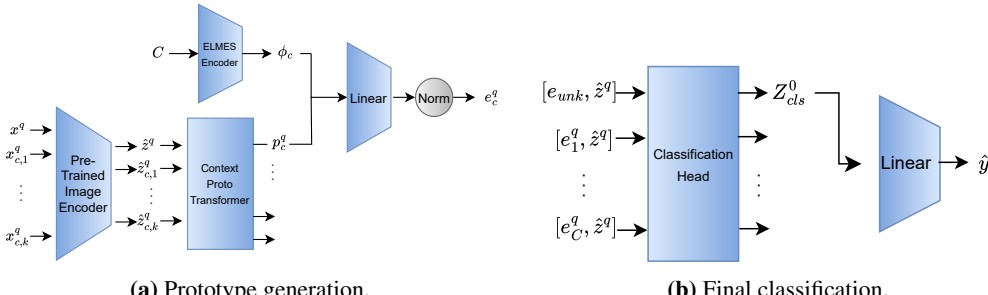

**(a)** Prototype generation.  **(b)** Final classification.

**Figure 1:** Method overview. (a) Prototype generation for class $c \in \{1, \ldots, C\}$ given query image $x^q$. The top-$k$ most similar training samples $\{x_{c,1}^q, \ldots, x_{c,k}^q\}$ from class $c$ are selected by cosine similarity. All images are encoded into normalized embeddings $\hat{z}^q$ and $\{\hat{z}_{c,j}^q\}_{j=1}^k$, fused via a transformer to yield the query-conditioned prototype $p_c^q$. Separately, class index $c$ is embedded via the fixed ELMES encoder into $\phi_c$. Both are combined and projected to produce the final query-aware prototype $e_c^q$, used for classification. (b) Final classification using contextual prototypes. The query is concatenated with each $e_c^q$ and passed through a transformer. The 'unknown' token output is projected to predict the final label.

examples, without fusing the query, and learn distance metrics to classify queries. In contrast, our method selects support examples from the training set for each class, based on their visual similarity to the query, and uses a transformer to fuse them with the query to generate query-aware prototypes.

MAML (Finn et al., 2017) learns to quickly adapt an initialization through a few gradient steps in novel tasks. Transductive label propagation methods like TPN (Liu et al., 2019) build a graph over support and query examples to diffuse labels throughout the episode. More recent transformer-based meta-learners, including CAML (Fifty et al., 2024b) and GPICL (Kirsch et al., 2022), jointly encode support and query sets in a single pass, enabling rich context modeling without iterative fine-tuning. Similarly, the proposed method uses transformers to encode support-query interactions. However, unlike previous work, our method constructs a tailored prototype for each query early in the pipeline, which improves both contextualization and computational efficiency.

Dynamic few-shot methods generate classification weights on-the-fly (Gidaris & Komodakis, 2018) or adapt embeddings via set-to-set transformers (Ye et al., 2020), while MetaOptNet (Lee et al., 2019) aligns features with a differentiable SVM solver for robust linear classification.

CAML (Fifty et al., 2024b) frames few-shot classification as non-causal sequence modeling over support-query pairs, using a frozen image encoder and a transformer. Its main contribution is the introduction of ELMES label embeddings, a fixed, equal-length, maximally equiangular vectors, which preserve permutation invariance and stabilize class identity encoding. This allows CAML to learn new classes at inference without fine-tuning, achieving state-of-the-art results across diverse benchmarks. Following CAML, our method adopts ELMES label embeddings to provide structured, geometry-aware class identity signals. However, there are important differences. CAML jointly encodes the entire support and query set in a single transformer to model task-level context and does not construct explicit per-class prototypes - it relies on fixed, uniformly sampled supports. In contrast, our method performs query-guided support retrieval, selecting the most relevant examples per class, and generates query-conditioned prototypes via targeted fusion. This early fusion allows the model to focus on a compact set of class prototypes rather than full support-query sequences. As a result, it achieves improved performance and lower memory consumption while yielding more context-aware and discriminative representations.

## 3 METHOD

Figures 1a and 1b provide an overview of the full method. Figure 1a depicts the contextual prototype generation process, where the top-$k$ most similar support examples are selected and fused with the query to form a query-aware prototype. Figure 1b then presents the final classification step using these contextual prototypes. Training and optimization details are provided in Appendix A.1.

## 3.1 SUPPORT SET CONSTRUCTION

We follow a pipeline to construct a class-wise support set for each query sample. This process involves embedding extraction, normalization, and similarity-based retrieval from the training set.

**Embedding extraction.** Let $\mathcal{D}_{\text{train}} = \{(x_i, y_i)\}_{i=1}^{N_{\text{train}}}$ denote the training set, where $x_i \in \mathbb{R}^{H \times W \times C}$ is an image and $y_i \in \mathcal{Y}$ is its class label, for a total of $N_{\text{train}}$ training samples. Each image is passed through a pre-trained image encoder $f : \mathbb{R}^{H \times W \times C} \to \mathbb{R}^d$ to obtain a feature embedding and then normalize using L2 normalization:

$$\mathbf{z}_i = f(x_i) \tag{1}$$

$$\hat{\mathbf{z}}_i = \frac{\mathbf{z}_i}{\|\mathbf{z}_i\|_2} \tag{2}$$

The collection of normalized training embeddings and their labels is denoted by:

$$\mathscr{Z}_{\text{train}} = \{(\hat{\mathbf{z}}_i, y_i) \mid i = 1, \dots, N_{\text{train}}\} \tag{3}$$

**Query-specific support set retrieval.** Given a query image $x^q$, which may originate from the training or the test set, we compute its normalized embedding $\hat{\mathbf{z}}^q$ using the same procedure as in Equations equation 1 and equation 2.

For each class $c \in \mathcal{Y}$, we identify the top-$k$ most similar training examples from class $c$ based on cosine similarity. Specifically, we retrieve the $k$ highest-scoring embeddings from $\mathscr{Z}_{\text{train}}$, excluding the query image itself if it appears in the training set:

$$\mathcal{S}_c^q = \text{Top-}k\,(\hat{\mathbf{z}}^q, c) \tag{4}$$

where $\text{Top-k}\,(\hat{\mathbf{z}}^q, c)$ selects the $k$ training examples from class $c$ with the highest cosine similarity to the query embedding $\hat{\mathbf{z}}^q$. If the query comes from the training set, it is excluded from the candidate pool to prevent self-matching.

The complete support set for the query is formed by concatenating the class-specific selections:

$$\mathcal{S}^q = \bigcup_{c \in \mathcal{Y}} \mathcal{S}_c^q \tag{5}$$

Thus, $\mathcal{S}^q$ contains $k \times |\mathcal{Y}|$ training examples in total. Importantly, all supports are drawn exclusively from the training set $\mathcal{D}_{\text{train}}$, even when the query image is from the test set. This ensures consistent and fair retrieval and evaluation throughout training and testing.

## 3.2 ELMES LABEL EMBEDDING

To represent class identity in a geometrically balanced way, we use the equal length and maximum equiangular set (ELMES) (Fifty et al., 2024a), a non-learned set of vectors, to encode class labels.

For each class $c \in \mathcal{C}$, ELMES define a fixed label embedding $\phi_c \in \mathbb{R}^{d_\ell}$ satisfying:

$$\|\phi_c\| = \alpha, \quad \langle \phi_i, \phi_j \rangle = -\frac{\alpha^2}{|\mathcal{C}| - 1} \quad \forall i \neq j \tag{6}$$

This corresponds to the vertices of a regular simplex centered at the origin, where $\alpha$ is a parameter of ELMES. Following Fifty et al. (2024a), this parameter is set to 1.32 thoroughout all our experiments. The resulting ELMES structure maximizes the angular separation between labels while maintaining an equal norm, which promotes permutation invariance and minimizes the classification entropy within the attention-based classifier.

The complete matrix of embeddings of the label is denoted $\Phi \in \mathbb{R}^{C \times d_\ell}$, and a learnable vector $\phi_{\text{unk}} \in \mathbb{R}^{d_\ell}$ is included to represent the position of the query during both training and inference, where the true label is unknown.

### 3.3 CONTEXTUAL PROTOTYPE GENERATION

Given a query image $x^q$ and a support set $\mathcal{S}^q = \{(x_{c,j}^q, y_c) \mid c = 1, \ldots, C;\ j = 1, \ldots, k\}$, with $k$ labeled support images per class, we compute embeddings using a frozen pre-trained encoder $f : \mathbb{R}^{H \times W \times C} \to \mathbb{R}^{d_f}$.

For class $c$, we build a sequence by stacking the query embedding with its $k$ class support vectors:

$$\mathbf{Z}_c^q = \left[ \hat{\mathbf{z}}^q; \hat{\mathbf{z}}_{c,1}^q; \ldots; \hat{\mathbf{z}}_{c,k}^q \right] \in \mathbb{R}^{(k+1) \times d_f} , \tag{7}$$

where $\hat{\mathbf{z}}_{c,j}^q$ and $\hat{\mathbf{z}}^q$ are the normalized embeddings as defined in Equation 2.

This sequence is passed through a transformer encoder that we refer to as `ContextProtoNet`:

$$\mathbf{L}_c^q = \texttt{ContextProtoNet}(\mathbf{Z}_c^q) \in \mathbb{R}^{(k+1) \times d_f} , \tag{8}$$

Next, we use only the first output token of the transformer encoder as the class prototype:

$$\mathbf{p}_c^q = \mathbf{L}_c^q[0] \in \mathbb{R}^{d_f} , \tag{9}$$

Note that $\mathbf{p}_c^q$ reflects attention over the full input sequence, including the $k$ support embeddings. Note that although only the first output is used, the other tokens are critical for providing contextual information through self-attention and enabling the generation of a query-aware prototype.

### 3.4 CLASSIFICATION HEAD

Each contextual prototype $\mathbf{p}_c^q \in \mathbb{R}^{d_f}$ is fused with its corresponding embedding of the ELMES label $\phi_c \in \mathbb{R}^{d_\ell}$ by concatenation:

$$\mathbf{h}_c^q = [\mathbf{p}_c^q, \phi_c] \in \mathbb{R}^{d_f + d_\ell} \tag{10}$$

A projection layer $W_{\text{proj}} \in \mathbb{R}^{d_f \times (d_f + d_\ell)}$ maps the fused representation into a contextual space:

$$\mathbf{g}_c^q = W_{\text{proj}} \mathbf{h}_c^q \in \mathbb{R}^{d_f} \tag{11}$$

To improve geometric stability, we apply L2 normalization and a learned scaling factor $\gamma \in \mathbb{R}$:

$$\mathbf{e}_c^q = \gamma \cdot \frac{\mathbf{g}_c^q}{\|\mathbf{g}_c^q\|_2 + \varepsilon} \tag{12}$$

A learnable vector $\mathbf{e}_{\text{unk}} \in \mathbb{R}^{d_f}$ is used to represent the class 'unknown'.

Given a query embedding $\hat{\mathbf{z}}^q$, we construct the final input sequence to the classifier by concatenating it to each contextual class embedding along the feature dimension:

$$\mathbf{X}_{\text{cls}}^q = \begin{bmatrix} [\mathbf{e}_{\text{unk}}, \hat{\mathbf{z}}^q] \\ [\mathbf{e}_1^q, \hat{\mathbf{z}}^q] \\ \vdots \\ [\mathbf{e}_C^q, \hat{\mathbf{z}}^q] \end{bmatrix} \in \mathbb{R}^{(C+1) \times (2d_f)} \tag{13}$$

This sequence is passed into another Transformer encoder, and the output is a contextualized representation for each token:

$$\mathbf{C}_{\text{cls}} = \texttt{ClassificationHead}(\mathbf{X}_{\text{cls}}^q) \in \mathbb{R}^{(C+1) \times d_f} \tag{14}$$

Following the contextual prototype generation step, we use only the first token in the output sequence, which corresponds to the unknown/query position:

$$\mathbf{Z}_{\text{cls}} = \mathbf{C}_{\text{cls}}[0] \in \mathbb{R}^{d_f} \tag{15}$$

**Table 1:** Top-1 accuracy on CIFAR-100 (a) and ImageNet-1K (b) few-class subsets. All models are evaluated under the FCA protocol with identical class subsets. MT = Model Type (F: full model, FT: finetuned model, S: sub-model). SD = Std. Dev. over five subsets.

**(a)** CIFAR-100

| MT | $N_{CL}$ | ResNet18 | ResNet50 | MViT-S | ViT-B |
|---|---|---|---|---|---|
| F | 2 | 75.00 | 71.30 | 71.80 | 40.80 |
| FT | 2 | 87.90 | 93.70 | 90.50 | 95.20 |
| S | 2 | 96.30 | 95.30 | 95.50 | 95.90 |
| Ours | 2 | **98.20** | **97.40** | **96.20** | **100.00** |
| F | 4 | 75.10 | 72.20 | 72.35 | 36.15 |
| FT | 4 | 87.60 | 90.55 | 90.00 | 91.16 |
| S | 4 | 90.65 | 90.15 | 89.45 | 85.40 |
| Ours | 4 | **94.80** | **97.55** | **97.25** | **98.35** |

**(b)** ImageNet-1K

| MT | $N_{CL}$ | ResNet18 | | ResNet50 | | ViT-B | |
|---|---|---|---|---|---|---|---|
| | | Top-1 | SD | Top-1 | SD | Top-1 | SD |
| F | 2 | 62.80 | 14.18 | 70.60 | 14.67 | 78.00 | 14.47 |
| S | 2 | 96.80 | 1.304 | 94.80 | 2.168 | 95.60 | 1.949 |
| Ours | 2 | **99.80** | **0.447** | **100.00** | **0.000** | **100.00** | **0.000** |
| F | 5 | 65.68 | 9.680 | 72.16 | 9.302 | 79.12 | 8.146 |
| S | 5 | 93.68 | 1.213 | 92.96 | 3.170 | 89.60 | 2.227 |
| Ours | 5 | **98.88** | **0.769** | **99.28** | **0.716** | **99.44** | **0.456** |
| F | 10 | 66.68 | 4.372 | 73.76 | 3.675 | 79.52 | 2.704 |
| S | 10 | 91.88 | 1.640 | 91.48 | 2.265 | 82.16 | 3.508 |
| Ours | 10 | **97.68** | **0.559** | **98.80** | **0.748** | **99.24** | **0.434** |

**Table 2:** Top-1 accuracy on the Few-Class Arena benchmark, with $N_{CL} = 5$ labeled classes per dataset. FT = fine-tuned. All methods are evaluated under the same FCA protocol.

| Model | Caltech 101 | Caltech 256 | CIFAR 100 | CUB 200 | Food 101 | GTSRB 43 | ImageNet 1K | Indoor 67 | Textures 47 |
|---|---|---|---|---|---|---|---|---|---|
| FT Swin V2 (Liu et al., 2021) | 88.07 | 92.07 | 93.92 | 67.53 | 94.48 | 86.02 | 95.28 | 91.40 | 48.50 |
| FT CoAtNet (Dai et al., 2021) | 93.66 | 96.72 | 92.56 | 87.78 | 94.80 | 89.86 | 97.60 | 96.81 | 52.50 |
| FT ConvNeXt V2 (Woo et al., 2023) | 91.70 | 94.79 | 91.36 | 62.17 | 92.48 | 89.68 | 92.40 | 92.76 | 47.90 |
| CAML (Fifty et al., 2024a) | 89.40 | 92.46 | 72.98 | 89.64 | 87.16 | 57.52 | 94.76 | 92.70 | 71.40 |
| CAML + **SIM** | 92.78 | 93.04 | 81.96 | 91.12 | 95.47 | 59.52 | 96.80 | 96.57 | 78.50 |
| Ours Swin V2 | 99.50 | **98.85** | 96.94 | **98.52** | 97.54 | 95.75 | **99.96** | **98.21** | **91.60** |
| Ours CoAtNet | **99.71** | 98.58 | 96.16 | 97.39 | 97.28 | 99.52 | 99.55 | 95.16 | 83.35 |
| Ours ConvNeXt V2 | 99.66 | 98.05 | **97.16** | 93.02 | 97.07 | **99.74** | 99.13 | 96.08 | 79.50 |

Next, we apply a linear projection $W_{\text{out}} \in \mathbb{R}^{d_f \times C}$ to produce the class logits:

$$\boldsymbol{\pi} = \mathbf{Z}_{\text{cls}} \cdot W_{\text{out}} \in \mathbb{R}^C \tag{16}$$

where $\boldsymbol{\pi}$ contains scores for the $C$ known classes. The predicted label is:

$$\hat{y} = \arg\max_i \boldsymbol{\pi}_i \tag{17}$$

## 4 EXPERIMENTS

We evaluate our method under the FCA protocol of Cao et al. (2024), which includes baselines such as full models trained on all classes, fine-tuned models on subsets, and submodels trained from scratch. We also compare with recent few-shot methods. Experiments across diverse datasets, architectures, and class subset sizes demonstrate the robustness and generality of our approach.

### 4.1 DATASETS

We adopt the image classification datasets from the FCA benchmark (Cao et al., 2024), including CalTech101 (Fei-Fei et al., 2007), CalTech256 (Griffin et al., 2007), CIFAR-100 (Krizhevsky, 2009a), CUB-200 (Wah et al., 2011), Food-101 (Bossard et al., 2014), GTSRB (Stallkamp et al., 2012), ImageNet-1K (Deng et al., 2009a), Indoor67 (Quattoni & Torralba, 2009), and Textures-47 (Cimpoi et al., 2014). These datasets span natural objects, food, fine-grained species, traffic signs, indoor scenes, and textures, representing real-world few-class use cases.

To ensure comparability with FCA, we use the same random seeds to generate class subsets and follow their protocol for evaluating each dataset at $N_{\text{CL}} \in 2, 4, 5, 10$, with classes sampled uniformly without replacement. All reported means and standard deviations are averaged over five such subsets. For fairness, our choice of $N_{\text{CL}}$ values, metrics, and base models is dictated entirely by FCA.

### 4.2 EVALUATION UNDER FCA PROTOCOL

The Few-Class Arena (FCA) benchmark (Cao et al., 2024) defines three training strategies to evaluate the performance of the model in the few-class regime.

**The full model (F).**   Trained once on all dataset classes and evaluated on few-class subsets without adaptation. Despite its efficiency, it often underperforms in few-class settings due to task mismatch and limited target classes.

**The fine-tuned model (FT).**   Starts with a full model and adapts it to the few-class target set. ResNet-18 and ResNet-50 are fine-tuned for 20 epochs, while ViT-Base, initialized from a CLIP-pretrained backbone, uses a linear head trained for 10 epochs. This improves alignment with the target task but still depends on the original full-model training.

**The sub-model (S).**   Trained from scratch on a subset of classes sampled from the full dataset. Each is independently optimized to convergence. While this yields strong tailored performance, it requires retraining for every new class configuration, incurring high computational cost.

**Our approach.**   We avoid the limitations of the baselines by keeping the ImageNet-1K-pretrained backbone frozen, and training only a lightweight transformer head from scratch for each few-class problem. This head builds query-guided prototypes from a few support images per class, adding only a few million parameters that converge in minutes on a single GPU. The design delivers better performance, requires no access to full-dataset labels, and scales easily to any class subset, making it a simple, fast, and flexible solution for real-world few-class deployment.

In Table 1a, we show a comparison of our method with the three FCA baselines on CIFAR-100 (Krizhevsky, 2009b), using subsets of $N_{CL} \in \{2, 4\}$ classes. All methods are evaluated under the same protocol, with our approach using five support images per class selected via query-guided retrieval. Across both settings and all architectures, our method consistently outperforms the full, fine-tuned, and sub-model baselines. Unlike **F**, which is trained on the full label space, **FT**, which requires adaptation from it, and **S**, which retrains a new model for each class subset, our method operates directly in the few-class setting using a frozen backbone and a lightweight, query-guided head. The observed improvements are consistent across both lightweight and large backbones, indicating that performance gains arise from adaptive support selection and decision boundary refinement rather than model size.

To better assess the robustness of our method, we evaluated on ImageNet-1K (Deng et al., 2009b) using the FCA (Cao et al., 2024) protocol with subsets of $N_{CL} \in \{2, 5, 10\}$ classes. Experiments are conducted on ResNet-18, ResNet-50, and ViT-Base, and results are compared to **F** and **S**, where the latter represents the strongest configuration in the original benchmark. As shown in Table 1b, our method consistently outperforms all baselines in all architectures and class counts.

Performance gains are observed not only in moderate settings with few classes, but also in the most constrained case of $N_{CL} = 2$, where our method achieves a near-perfect classification. These improvements are accompanied by low variance across trials and hold across lightweight and large backbones. The results show that even strong feature extractors benefit from the fine-grained adaptation of our method, reinforcing the conclusions from CIFAR-100.

Table 2 shows top-1 accuracy on nine datasets with $N_{CL} = 5$ classes using Swin V2 (Liu et al., 2021), CoAtNet (Dai et al., 2021), and ConvNeXt V2 (Woo et al., 2023). We also compare with CAML (Fifty et al., 2024a), a leading few-shot meta-learning model. Our method consistently outperforms all baselines. While CAML encodes the full support-query sequence to model task context, it uses fixed, uniformly sampled supports and lacks explicit per-class prototypes. In contrast, our method retrieves top-$k$ relevant supports per query, performs early fusion to generate query-conditioned prototypes, and attends to compact, class-specific representations. This improves accuracy, reduces memory usage, and yields more discriminative features. We also evaluate CAML+SIM, which adds similarity-based support selection; though it slightly improves CAML's accuracy, it still falls short of our query-conditioned prototypes, highlighting the importance of targeted fusion.

### 4.3 Performance Analysis Across Backbone Sizes

To assess robustness across model scales and datasets, we analyze ResNet backbones of increasing capacity (18, 34, 50, 101, 152) on the nine FCA (Cao et al., 2024) datasets, each evaluated with five support images per class. Results are shown in Figure 2.

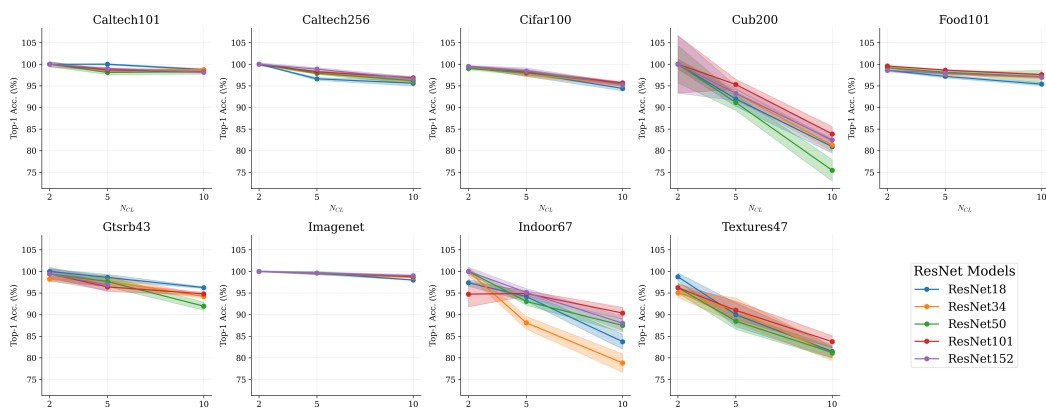

**Figure 2:** Classification accuracy (%) of ResNet backbones using our method as $N_{\text{CL}}$ increases from 2 to 10. Each subfigure shows a dataset. Shaded ribbons show $\pm 1$ SD over five random class selections, each with five support images per class.

As expected, Top-1 accuracy decreases as $N_{\text{CL}}$ increases from 2 to 10 across all datasets and backbones, reflecting the added complexity and decision boundaries needed to separate more classes.

Interestingly, accuracy is no longer strictly tied to size. In transfer learning, larger models (e.g., ResNet-152) typically outperform smaller ones (e.g., ResNet-18), but in the few-class regime with limited supervision, this relationship becomes dataset dependent and can even invert. On Caltech-101, Caltech-256, and CIFAR-100, larger ResNets maintain a slight edge, while on Cub-200, Indoor-67, and Textures-47, smaller models often match or outperform them. This suggests that in low-data settings, larger models may overfit or fail to fully utilize their capacity without sufficient support.

Cub-200, Indoor-67 and Textures-47 drop more steeply with increasing $N_{\text{CL}}$, likely due to greater intraclass variability and fine-grained class structure.

Standard deviation across seeds is lower in datasets with low intraclass variability (e.g., Caltech-101, Food-101, CIFAR-100). On ImageNet-1K subsets, Top-1 accuracy remains stable even as class count grows from 2 to 10 with 5 support images. These results confirm our method generalizes well across datasets and architectures, achieving high accuracy with minimal variance. They also show that larger backbones are not always optimal, and smaller or mid-sized models can outperform them when paired with effective support-based optimization, as in our approach.

Figure 2 and Table 1b show a unified trend: our method is largely invariant to backbone size, while FCA baselines degrade or become unstable as capacity increases. Across all ResNets, our accuracy curve remains flat, with tight shaded bands showing minimal run-to-run variation. This demonstrates that freezing the encoder and training only a lightweight, query-aware head prevents the overfitting large networks typically suffer in few-class settings. In contrast, the full-model (F) baseline gains little from deeper networks and shows increasing variance, while the sub-model (S) often peaks with the smallest backbone before plateauing or declining. As shown in Table 1b and Figure 2, our method consistently yields lower standard deviation across all backbones and class counts, whereas F and S exhibit much higher variability under identical conditions.

Overall, our method consistently achieves higher accuracy and greater robustness than all baselines, regardless of model size, dataset, or number of classes.

### 4.4 ABLATION ANALYSIS

To identify the sources of our gains, we ablate two components: (1) **Query fusion** - during prototype generation, the query is fused with its supports to produce query-conditioned prototypes (QC). When removed for ablation, the head reduces to a plain class centroid, i.e., the uniform average of the $k$ support features; (2) **Support selection** - we compare similarity-based selection (SIM), which selects the $k$ nearest neighbors of the query in each class, with random selection (RAND), which samples supports uniformly.

**Table 3:** Top-1 accuracy and convergence epochs for frozen Swin V2-Tiny across four ablations. **QC**: query-conditioned prototype or using the plain average of the support features; **Support**: selecting support examples randomly (RAND) or by similarity to the query (SIM). All use $N_{CL} = 5$ with five supports per class.

| Dataset | QC | Support | Acc | Epochs |
|---|---|---|---|---|
| CUB-200 | ✗ | RAND | 78.50 | 15 |
| | ✗ | SIM | 86.97 | 13 |
| | ✓ | RAND | 98.52 | 7 |
| | ✓ | SIM | 98.79 | 5 |
| Indoor-67 | ✗ | RAND | 72.36 | 19 |
| | ✗ | SIM | 92.99 | 13 |
| | ✓ | RAND | 98.21 | 5 |
| | ✓ | SIM | 99.00 | 4 |
| Textures-47 | ✗ | RAND | 69.12 | 17 |
| | ✗ | SIM | 81.90 | 11 |
| | ✓ | RAND | 90.59 | 4 |
| | ✓ | SIM | 91.90 | 3 |

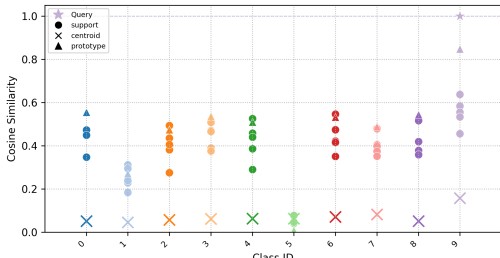

**Figure 3:** Cosine similarity between the query (star, class ID 9), support images (solid circles), plain centroids (cross), and query-conditioned prototypes (triangles) for each class in a 10-way, 5-shot CUB-200 episode. The correct-class QC prototype is closest to the query, highlighting its superiority over uninformed centroids.

In Table 3 we evaluate all four combinations (QC $\times$ SIM/RAND) under the FCA protocol with $N_{CL} = 5$ and five supports per class, using a frozen Swin-V2 Tiny backbone trained for 20 epochs on CUB-200, Indoor-67, and Textures-47. We report Top-1 accuracy and the epoch where the running mean stabilizes (variation $< 0.1\%$ over three checkpoints).

The results reveals three trends: **(i)** QC is the key factor, markedly improving accuracy across support strategies; **(ii)** Without QC, support selection is crucial - SIM outperforms RAND, highlighting the value of informative prototypes, while with QC, fusion mitigates weak supports; **(iii)** QC dramatically reduces convergence time, indicating that early query integration accelerates learning.

Appendix A.2 presents results for different numbers of retrieved support images per class.

**Prototype Geometry** To illustrate how query-conditioned fusion reshapes prototypes, we examine a ten-way, five-shot episode from CUB-200. In Figure 3 we compute for each class, the cosine similarity between the query (purple star) and three references: (1) the five support embeddings (solid circles), (2) the centroid computed by averaging supports uniformly (cross), and (3) the query-conditioned fused prototype (triangle).

Across all classes, the QC prototype is consistently closest to the query, while the plain centroid often drifts, especially with dispersed supports. This aligns with Table 3, showing that QC pulls prototypes toward the query, reducing bias, while SIM provides cleaner support examples. Together, they explain why the QC+SIM setup achieves superior accuracy and faster convergence.

Overall, query-conditioned fusion and targeted support selection are complementary: the former tailors prototypes to each query, and the latter ensures informative inputs, yielding both, higher accuracy and more stable and efficient training in the few-class regime.

## 5 CONCLUSION

We introduced a query-aware classification head that dynamically retrieves and fuses support examples to generate query-conditioned prototypes. Evaluated across nine datasets, our method consistently outperforms strong baselines, demonstrating superior accuracy and robustness in the few-class setting. Its lightweight design and compatibility with frozen encoders make it practical for real-world deployment across diverse architectures and tasks.

USE OF LARGE LANGUAGE MODELS

ChatGPT was used only to aid and polish the writing of this paper.

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

# A  APPENDIX

## A.1  IMPLEMENTATION DETAILS

We use a frozen SwinV2-Tiny backbone with input size $224 \times 224$ and drop path rate 0.2, serving purely as a feature extractor. Training is applied only to the additional components: the `ContextProtoNet` transformer for query-aware prototype generation, the projection layer $W_{proj}$ that fuses each prototype with its fixed ELMES label embedding, the learnable unknown class embedding $\mathbf{e}_{unk}$, the scaling factor $\gamma$ applied after L2 normalization, and the `ClassificationHead` transformer including the output matrix $W_{out}$.

We use the AdamW optimizer with $\beta_1 = 0.9$, $\beta_2 = 0.999$, $\varepsilon = 10^{-8}$, and weight decay $5 \times 10^{-2}$, along with gradient clipping at maximum norm 5. A single-cycle cosine annealing schedule is used, starting with a warmup learning rate of $1 \times 10^{-4}$ and decaying to $1 \times 10^{-5}$ over 30 epochs. The first 5 epochs use linear decay, followed by cosine decay. For each class, we retrieve the top $k = 5$ most similar training samples by cosine similarity to form the support set. Models are trained for 30 epochs with batch size 48 using cross-entropy loss.

All experiments were conducted using PyTorch on an NVIDIA RTX A6000 GPU.

## A.2  EFFECT OF $k$ RETRIEVED SUPPORTS

Table 4 shows the effect of varying the number of retrieved support images per class ($k \in \{1, 3, 5, 9\}$) on ResNet-50, averaged over the nine datasets presented in Table 2 and $N_{CL} \in \{2, 4, 10\}$. Even with $k = 1$, our method outperforms the fine-tuned baseline. Accuracy improves with larger $k$, saturating around $k = 5$, while FLOPs and latency continue to increase. We adopt $k = 5$ as the default, balancing performance and efficiency.

**Table 4:** Effect of class supports on ResNet-50, averaged over nine datasets and $N_{\mathrm{CL}} \in \{2, 4, 10\}$.

| $k$ | 1 | 3 | 5 | 9 |
|---|---|---|---|---|
| Top-1 (%) | 96.3 | 97.4 | **98.1** | 98.1 |
| $\Delta$FLOPs | +1.7% | +2.8% | +4.8% | +8.6% |
| Latency (ms) | 1.9 | 2.1 | 2.5 | 3.4 |

