# OpenReview forum: "Query-Guided Prototype Generation for Few-Class Classification"
_ICLR.cc/2026/Conference — ICLR 2026 Conference Withdrawn Submission_

### Official Review · Reviewer_JFPj · 2025-10-26

**Soundness:** 2
**Presentation:** 2
**Contribution:** 2
**Rating:** 4
**Confidence:** 4

**Summary:**

This paper investigates the few-class regime problem. The authors propose a new few-shot class classification framework that generates prototypes by retrieving and fusing support samples guided by the query. The proposed method achieves state-of-the-art performance across multiple few-class classification benchmarks.

**Strengths:**

1. The few-class regime is an interesting and practically research problem due to its real-world applicability.
2. The proposed method is clearly presented and achieves state-of-the-art performance across benchmarks.

**Weaknesses:**

1. The motivation is not clear. It seems that the authors propose an incremental few-shot learning method for the few-class classification. The authors should provide a more explicit and detailed explanation of the motivation behind the proposed method.
2. A clear definition and formulation of the problem should be provided before the introduction of the method.
3. The authors should add relevant references in the introduction (lines 44-45).
4. In lines 145–152, the authors emphasize their differences from CAML; however, the innovation appears rather intuitive and limited in novelty.
5. According to Table 3, the performance gain from support selection becomes marginal after applying query fusion. Nevertheless, the authors emphasize this component as a key contribution.
6. Table 2 does not include the results for the sub-model (S).

**Questions:**

1. The structure of ContextProtoNet is not clearly described. How significantly do different design choices in this component affect the overall performance?
2. Could the proposed approach be generalized beyond classification to other vision tasks such as semantic segmentation?

---

### Official Review · Reviewer_RBds · 2025-10-29

**Soundness:** 2
**Presentation:** 3
**Contribution:** 2
**Rating:** 2
**Confidence:** 4

**Summary:**

This paper presents a novel framework for few-shot learning that introduces a query-aware classification head. The key innovation is the dynamic retrieval and fusion of the most relevant support examples to form query-conditioned prototypes, moving beyond static support sets. Extensive experiments on the Few-Class Arena benchmark demonstrate that this approach achieves state-of-the-art performance, signifying a robust and effective advancement for tasks with limited class data.

**Strengths:**

1.The paper's most significant innovation is replacing the common practice of using a randomly sampled and query-agnostic support set with a dynamic method that retrieves the most relevant support samples from the training set for each query to generate subsequent prototypes.

2.The paper is well-written with a clear structure and supported by comprehensive experiments.

**Weaknesses:**

1. The current title does not adequately highlight the paper's core innovation—the novel dynamic support set construction paradigm. A more descriptive title could more effectively signal this contribution to the reader.

2. While the dynamic support retrieval is innovative, the subsequent steps for generating query-aware prototypes heavily rely on established techniques, specifically the ELMES strategy from CAML and a standard Transformer encoder. The paper would be strengthened by clarifying what specific novelty is introduced in this fusion stage beyond the retrieval mechanism itself.

3. The idea of generating query-aware representations has precedents in few-shot learning literature that are not sufficiently discussed. For instance, the CTX[1] model also leverages Transformer-based feature alignment between support and query sets. The lack of a direct comparison with such relevant approaches in the Related Work and Experiments sections makes it difficult to precisely gauge the incremental contribution of the proposed method over this existing concept.

Reference：

[1] Doersch C, Gupta A, Zisserman A. Crosstransformers: spatially-aware few-shot transfer[J]. Advances in Neural Information Processing Systems, 2020, 33: 21981-21993.

**Questions:**

1.Regarding Section 3.1, is the number of training samples, N_train, defaulted to using all available training samples per class from the dataset? Also, are both the training and test sets derived entirely from the dataset of novel classes?

2.During the stage of retrieving support examples, how is the specific scenario handled when the query sample itself comes from the training set?

3.The strategy of dynamically retrieving a support set for each query sample increases the temporal and spatial complexity. Does the paper need to discuss this aspect?

4.Why is it necessary to add a discussion of related work from biological pathology in the Related Work section?

---

### Official Review · Reviewer_8SUA · 2025-10-31

**Soundness:** 2
**Presentation:** 2
**Contribution:** 1
**Rating:** 2
**Confidence:** 5

**Summary:**

The paper builds a query-guided head for few-class classification: for each query, it retrieves top-k training examples per class, fuses query+supports with a small transformer to make query-conditioned prototypes, mixes in label embeddings (ELMES), then a second transformer predicts the class while the backbone stays frozen. Claimed benefits: better accuracy, faster, and lighter than fine-tuning or meta-learning across many FCA datasets.

**Strengths:**

•	Idea is simple and modular: per-query retrieval + small transformers on top of frozen features; easy to plug in.
•	Good ablations (SIM vs QC) that show where gains come from. QC seems to matter most.
•	Broad evaluation across multiple datasets/backbones with consistent gains (as reported).

**Weaknesses:**

•	Limited Novelty: The paper frames “query-conditioned prototypes” as new, but similar query/episode-conditioned prototype ideas exist, please reposition and cite accordingly.
•	Baselines incomplete: Missing strong, standard comparisons (e.g., FEAT/MetaOptNet) and a clean linear-probe/logistic-reg head on the same frozen features.
•	Some results feel suspicious: ImageNet-1K NCL=2 shows 100.0% with 0.0 SD in places. This needs per-subset class lists and a sanity check.
•	Efficiency claims ignore retrieval. Paper doesn’t report index type, build time, or retrieval cost in the timing.
•	Limited details for reproducibility: Need exact splits/seeds for FCA, full configs for every backbone, CAML settings, and sensitivity for ELMES α.
•	Potential leakage/duplicates: With retrieval from the train set, please discuss overlap handling to avoid accidental leakage, specially in scene datasets.

**Questions:**

see above weaknesses

---

### Official Review · Reviewer_bRrs · 2025-11-01

**Soundness:** 2
**Presentation:** 2
**Contribution:** 3
**Rating:** 2
**Confidence:** 4

**Summary:**

This paper proposed a novel prototype generation method for few-class classification tasks. Authors introduce a query-guided support retrieval and fusion pipeline, which trains a lightweight transformer head and classification head for downstream few-class learning. This study finds that prototypes generated from individual queries are more effective than simply aggregating support samples. Additionally, top-k sample retrieval further enhances few-class classification performance while reducing memory and computational overhead.

**Strengths:**

1. This paper focuses on investigating the impact of query-guided prototypes on few-class classification tasks, which is interesting and worth further exploration.

2. The experimental results and conclusions drawn in the paper partially support the authors' claims, while the proposed method is straightforward and easy to follow.

**Weaknesses:**

1. Authors argue that query-guided prototype generation is the key contribution of this paper, which leads to the most performance gain of the proposed method. However, some modules have also been introduced in this paper, such as the ELMES encoder, and ablation studies for these components are missing.

2. The experiments are insufficient. The baselines selected by authors are too weak, and the compared methods within the same research field are too few. More recent few-class learning methods from the past two years should be incorporated for comparison.

3. Authors use substantial space to elaborate on the computation overflow of each module in the pipeline, which appears somewhat redundant. Most of this kind of information is already illustrated in Fig. 1. Authors should instead supplement the discussion with insights into why the query-guided prototype generation is effective.

**Questions:**

1. The setup of the ablation experiments is somewhat confused. In Table 3, when using the plain average of support features (where QC is X), it is unclear whether the ELMES encoder is used. Moreover, how are the averaged features, query embeddings, and label embeddings combined?

2. It seems that SIM (the top-k retrieval) is less effective when applying QC, but it significantly improves the performance when using the average of support features. Is there a rational explanation? Could it be inferred that the benefits of QC primarily stem from the re-learning of the query (through the training of the transformer head and classification head, and the application of ELMES), rather than from learning from the retrieved similar training samples?

3. Top-k sample retrieval is a commonly used and intuitive method for generating query-specific prototypes. Its application to enhance prototype generation is hardly surprising. Have the authors considered whether the trivial cosine similarity serves as a well-suited distance metric for retrieval?

4. Instead of using the plain average of support features, have the authors considered concatenating the support features as prototypes?

5. It is confusing why the FT experiments are missing in Table 1 (b).

6. The ELMES does not seem appropriate for query-specific prototypes, since it constructs the latent space by maximizing angular separation between label embeddings. Authors should discuss the rationale for applying ELMES in query-guided prototype generation.

---

### Note · Authors · 2025-11-19

I have read and agree with the venue's withdrawal policy on behalf of myself and my co-authors.